# Organization and function of *Drosophila* odorant binding proteins

**Nikki K Larter[1,2], Jennifer S Sun[1], John R Carlson[1,2]\***

[1]Department of Molecular, Cellular and Developmental Biology, Yale University, New Haven, United States; [2]Interdepartmental Neuroscience Program, Yale University, New Haven, United States

**Abstract** Odorant binding proteins (Obps) are remarkable in their number, diversity, and abundance, yet their role in olfactory coding remains unclear. They are widely believed to be required for transporting hydrophobic odorants through an aqueous lymph to odorant receptors. We construct a map of the *Drosophila* antenna, in which the abundant Obps are mapped to olfactory sensilla with defined functions. The results lay a foundation for an incisive analysis of Obp function. The map identifies a sensillum type that contains a single abundant *Obp*, *Obp28a.* Surprisingly, deletion of the sole abundant *Obp* in these sensilla does not reduce the magnitude of their olfactory responses. The results suggest that this Obp is not required for odorant transport and that this sensillum does not require an abundant Obp. The results further suggest a novel role for this Obp in buffering changes in the odor environment, perhaps providing a molecular form of gain control.

## Introduction

Like many animals, insects rely on their sense of smell to navigate through the environment towards food sources and mates. The *Drosophila* antenna is covered with olfactory sensilla that fall into three main morphological classes: basiconic, trichoid and coeloconic sensilla (*Figure 1A,B*) (*Shanbhag et al., 1999*). Basiconic sensilla detect many fruit odors (*de Bruyne et al., 2001*; *Hallem and Carlson, 2006*; *Hallem et al., 2004*); trichoid sensilla sense pheromones (*Clyne et al., 1997*; *Dweck et al., 2015*; *Ha and Smith, 2006*; *van der Goes van Naters and Carlson, 2007*); coeloconic sensilla detect organic acids and amines (*Abuin et al., 2011*; *Ai et al., 2010*; *Benton et al., 2009*; *Silbering et al., 2011*; *Yao et al., 2005*). Each class can in turn be divided into functional types. For example, 10 types of basiconic sensilla, designated ab1 (antennal basiconic 1) through ab10, detect different subsets of odorants (*Couto et al., 2005*; *de Bruyne et al., 2001*; *Elmore et al., 2003*; *Hallem et al., 2004*; *Marshall et al., 2010*). Olfactory sensilla are perforated by pores or channels through which odorants can pass, and they contain an aqueous lymph in which the dendrites of up to four olfactory receptor neurons (ORNs) are bathed (*Figure 1C*). Many sensilla contain two ORNs, designated A and B based on their spike amplitudes. Sensilla also contain a thecogen (sheath) cell, a trichogen (shaft) cell, and one or two tormogen (socket) cells. These cells produce the lymph and wrap around the ORNs (*Shanbhag et al., 2000*).

Two broad classes of proteins are believed essential for the responses of sensilla to odorants. One class, the odor receptors, have been mapped to specific types of sensilla and to individual neurons within those sensilla, and their functional specificities have been characterized (*Ai et al., 2010*; *Benton et al., 2009*; *Couto et al., 2005*; *Hallem and Carlson, 2006*; *Hallem et al., 2004*; *Silbering et al., 2011*). The other class, called odorant binding proteins (Obps), have been the subject of much investigation in many insects but remain poorly understood (*Leal, 2013*; *Pelosi et al., 2006*; *Vogt and Riddiford, 1981*).

**\*For correspondence:** john. carlson@yale.edu

**Competing interests:** The authors declare that no competing interests exist.

**eLife digest** Insects use their sense of smell to find mates, to find food and – in the case of insects that transmit diseases such as malaria and Zika – to find us. If we can understand how insect scent detection works at the molecular and cellular level, we may be able to devise new ways of manipulating the insects' sense of smell and prevent them from finding us.

Insects contain a family of proteins called odorant binding proteins that are intriguing in several ways. They are numerous (there are 52 kinds in the fruit fly *Drosophila*), they are diverse and some are made in remarkably large amounts in the antennae. Fine hair-like structures known as olfactory sensilla protrude from the surface of the antennae. Odorant binding proteins are widely believed to carry odorant molecules through the fluid inside the sensilla to olfactory neurons, which then send signals that trigger the insect's response to the scent.

Larter et al. have now mapped the most abundant odorant binding proteins to the various olfactory sensilla of *Drosophila*. This revealed that a type of sensillum known as ab8 contained only one abundant odorant binding protein, called Obp28a. Unexpectedly, Larter et al. found that ab8 sensilla that are deprived of this protein respond strongly to odorant molecules. This result suggests that Obp28a is not required to transport odorants to the neurons in ab8; indeed, it appears that these neurons do not require an abundant odorant binding protein in order to respond to a scent. Instead, Obp28a helps to moderate the effects of sudden changes in the level of an odorant in the environment, so that concentrated odors do not trigger too large a response from the olfactory neurons.

The details of the role that Obp28a plays in olfactory sensilla remain to be investigated in future studies, and the map created by Larter et al. also lays a foundation for studying the roles of other odorant binding proteins. The discovery that Obp28a is not needed to transport odorant molecules also raises questions about how insects are able to detect smells. Many odorant molecules repel water, so how do these molecules travel through the fluid in the sensilla if odorant binding proteins are not needed to transport them?

Obps are remarkable in three ways: they are numerous, being encoded by a family of 52 genes in *Drosophila*; they are abundant, with some encoded by the most abundant mRNAs in the antenna; they are diverse, with members sharing only 20% amino acid identity on average (*Hekmat-Scafe et al., 2002*; *Menuz et al., 2014*). They are small proteins, on the order of 14 kDa, and despite their high sequence divergence they are believed to have a common structure (*Graham and Davies, 2002*). Many are found in the lymph of olfactory sensilla, where ORN dendrites are located (*Shanbhag et al., 2001a*). They bind odorants, with different degrees of affinity and selectivity reported for different Obps (*Gong et al., 2010*; *Leal et al., 2005*). Within a species, different Obps are expressed in different antennal sensilla (*McKenna et al., 1994*; *Pikielny et al., 1994*; *Schultze et al., 2013*), and some are expressed in the taste system (*Galindo and Smith, 2001*; *Jeong et al., 2013*; *Pikielny et al., 1994*; *Shanbhag et al., 2001b*) or in larval chemosensory organs (*Galindo and Smith, 2001*; *Park et al., 2000*).

Despite numerous studies of Obps, much remains to be learned about their function. They are widely believed to bind, solubilize, and transport hydrophobic odorants across the aqueous sensillum lymph to receptors in the dendrites (*Gomez-Diaz et al., 2013*; *Sandler et al., 2000*; *Vogt et al., 1985*; *Wojtasek and Leal, 1999*; *Xu et al., 2005*). Obps have also been proposed to accelerate the termination of odor response, by removing odorants from receptors or from the sensillar lymph (*Vogt and Riddiford, 1981*; *Ziegelberger, 1995*). A variety of studies support a role for Obps in olfactory perception in vivo (*Biessmann et al., 2010*; *Pelletier et al., 2010*; *Swarup et al., 2011*). However, to date, the physiological role of only one Obp, Obp76a, has been thoroughly investigated in the olfactory system of *Drosophila* (*Gomez-Diaz et al., 2013*; *Laughlin et al., 2008*; *Xu et al., 2005*). Obp76a, also called LUSH, is required in trichoid sensilla for normal response of the odor receptor Or67d to the pheromone cis-vaccenyl acetate (cVA), although responses of Or67d to cVA have been detected in the absence of Obp76a (*Benton et al., 2007*; *Gomez-Diaz et al., 2013*; *Li et al., 2014*; *van der Goes van Naters and Carlson, 2007*). LUSH has been found to bind

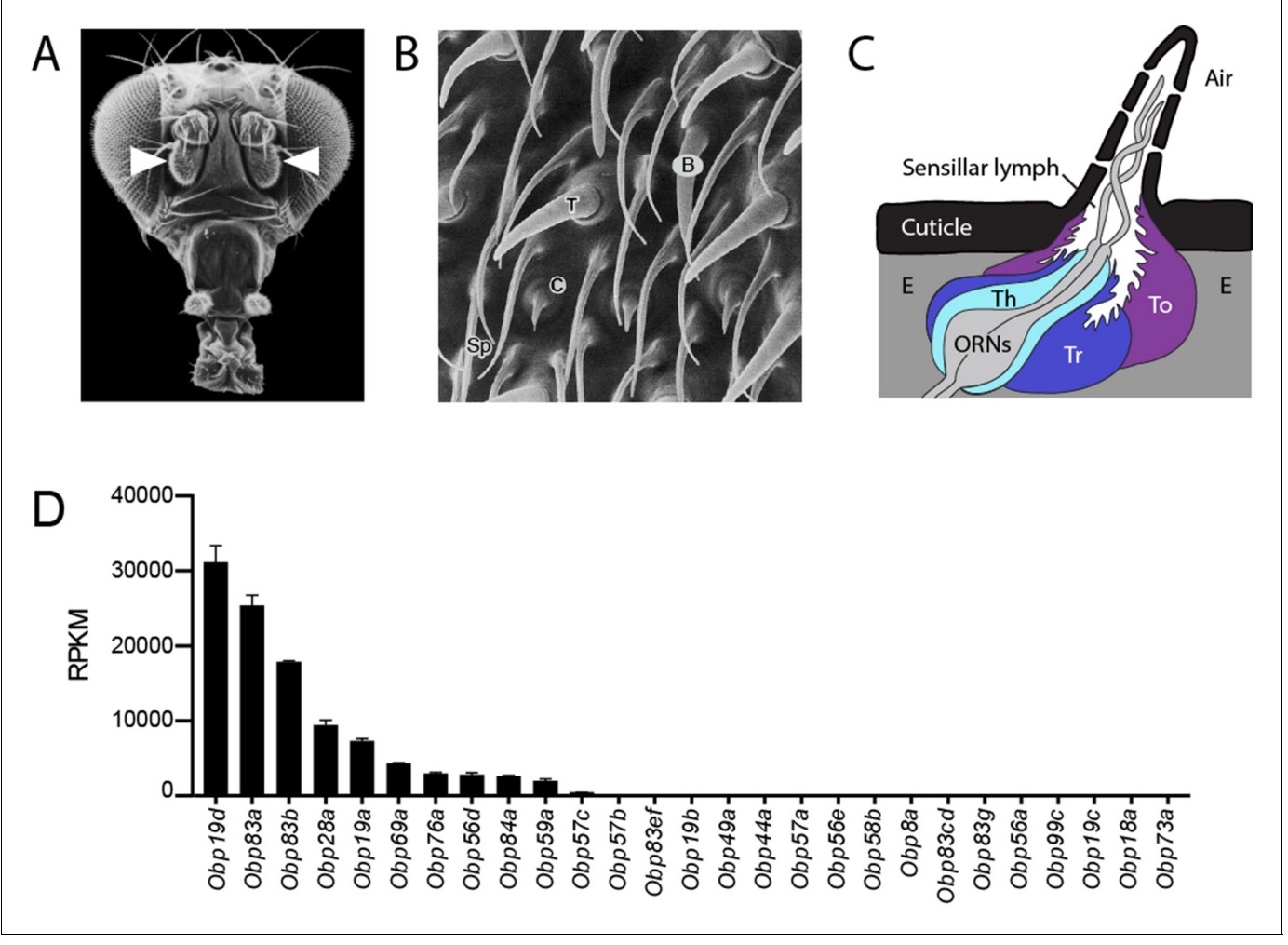

**Figure 1.** Organization of the antenna and expression of *Obps*. (A) Scanning electron micrograph of antennae (arrowheads) on a *Drosophila* head. Adapted from http://www.sdbonline.org/sites/fly/aimain/images.htm, (***Menuz et al., 2014***). (B) Higher magnification image of antennal surface showing basiconic (B), trichoid (T), and coeloconic (C) sensilla surrounded by non-innervated spinules (Sp). Adapted from (***Menuz et al., 2014***; ***Shanbhag et al., 1999***). (C) A generic sensillum containing two ORNs and thecogen (Th), trichogen (Tr), and tormogen (To) cells, separated from neighboring sensilla by epidermal cells (E). Adapted from (***Menuz et al., 2014***; ***Steinbrecht et al., 1992***). (D) Members of the *Obp* gene family detected in the third antennal segment, where olfactory sensilla are located, at >1 read per million mapped reads in each of three samples. Genes are listed in decreasing order of expression level, indicated in terms of reads per million mapped reads per kilobase of gene length (RPKM). Obp76a is also known as LUSH. Adapted from (***Menuz et al., 2014***).

The following figure supplement is available for figure 1:

**Figure supplement 1.** *Obp* expression levels.

cVA in vitro (***Kruse et al., 2003***; ***Laughlin et al., 2008***), but also binds other insect pheromones (***Katti et al., 2013***), short-chain alcohols (***Bucci et al., 2006***; ***Thode et al., 2008***), and phthalates (***Zhou et al., 2004***).

One reason the role of Obps has remained unclear is that their molecular organization has not been established. Previous work has not defined which Obps are expressed in individual sensilla, nor in which sensilla individual Obps are expressed. Thus it has been difficult to carry out well-defined manipulations of Obp content. Here we take advantage of a recent RNA-Seq analysis that quantified *Obp* expression in the antenna (***Menuz et al., 2014***). We examine the expression of the abundant *Obps* at the level of sensillum morphology, sensillum functional type, and cell type. The results illustrate basic principles of olfactory system organization, and they allow construction of an Obp-to-

sensillum map. The map identifies a sensillum type that contains only one highly expressed *Obp*, *Obp28a*. We delete this *Obp* gene and analyze the effects physiologically. Surprisingly, we find that deletion of the only abundant *Obp* in this sensillum type does not reduce the magnitude of its ORN responses. Additionally, we find evidence suggesting that this Obp plays a role not previously demonstrated for Obps, in buffering sudden increases in odorant levels. This Obp could provide a molecular mechanism of gain control that acts prior to receptor activation. Together, this work provides a foundation for incisive studies of Obp function, suggests that some sensilla do not require an abundant Obp for odorant transport, and encourages a broader view of the functions performed by the large and diverse family of Obps.

## Results

### Diverse expression patterns of abundant *Obps*

We systematically analyzed the expression patterns of the 10 *Obps* that are expressed most abundantly in the antenna. The remaining *Obps* are all expressed at much lower levels (*Figure 1D*, *Figure 1—figure supplement 1*). In situ hybridization was used to identify the regions of the antenna and the morphological classes of sensilla in which the abundant *Obps* are expressed.

These *Obps* are expressed in a wide diversity of spatial patterns (*Figure 2A*). Expression of some of the *Obps* was distributed broadly (*e.g. Obp19d*), expression of another was concentrated narrowly (*Obp59a*), and others showed intermediate patterns (*Obp76a*). The expression levels of these 10 *Obps* are striking not only in their magnitude but also in their wide range, from ~30,000 RPKM to ~2000 RPKM in the third antennal segment (*Figure 1D*) (*Menuz et al., 2014*). Consistent with the breadth of their spatial patterns, *Obp19d* is the most abundant transcript, and *Obp59a* is the least abundant of the 10 *Obps*.

Expression was observed in each of the three major morphological classes of sensilla (*Figure 2B*). Four, such as *Obp19a*, are expressed in basiconic sensilla; four, including *Obp76a*, are detected in trichoid sensilla; two, including *Obp84a*, are expressed in coeloconic sensilla (*Figure 2B,C*). Some *Obps* are expressed only in basiconic sensilla, some only in trichoid sensilla, and some only in coeloconic sensilla, but two *Obps* were expressed in both basiconic and trichoid sensilla, consistent with an earlier report (*Hekmat-Scafe et al., 1997*).

Of the two *Obps* in coeloconic sensilla, *Obp84a* is expressed in most if not all coeloconic sensilla on the antennal surface as well as those of the sacculus, a three-chambered cavity of the antenna. *Obp59a* was detected only in the sacculus. The localization of *Obp84a* and *Obp59a* to coeloconic sensilla is thus consistent with their absence in the antennae of *atonal*, a mutant lacking coeloconic sensilla, as revealed by an RNA-Seq analysis (*Menuz et al., 2014*).

*Obp19d* and *Obp56d* appear to be expressed not in olfactory sensilla but rather in epidermal cells, some of which flank olfactory sensilla and some of which are associated with uninnervated spinules (*Figure 2B*). *Obp56d* is also expressed in the arista, a feathery structure associated with thermosensation and mechanosensation (*Figure 2A*) (*Foelix et al., 1989*; *Ni et al., 2013*).

### An Obp-to-sensillum map of the *Drosophila* antenna

Having mapped the abundant *Obps* to morphological classes of sensilla, we next mapped them at higher resolution, to functional types of sensilla. We focused on the basiconic sensilla, whose function has been analyzed in particular detail (*de Bruyne et al., 2001*; *Hallem et al., 2004*). Our goal was to determine which of the abundant *Obps* are expressed in each of 10 individual functional types of basiconic sensilla.

We used 10 *Or-GAL4* drivers, each chosen to label a particular type of basiconic sensillum. For example, *Or42b-GAL4* labels an ORN located in the ab1 type, and *Or59b-GAL4* labels an ORN in the ab2 type. We systematically carried out a double-label analysis with 10 *Or-GAL4* drivers and in situ hybridization probes for each of the four *Obps* that are expressed in basiconic sensilla (*Figure 3*).

In this manner we found that ab1, one of whose ORNs is labeled green in the left column of *Figure 3A*, contained a cell labeled with *Obp83a* (red). Likewise, ab2, ab3, ab7, and ab10 all express *Obp83a*; in each case the dendrite of the labelled ORN in the sensillum appears to be surrounded by a cell expressing *Obp83a*. By contrast, the other five types of basiconic sensilla (ab4, ab5, ab6,

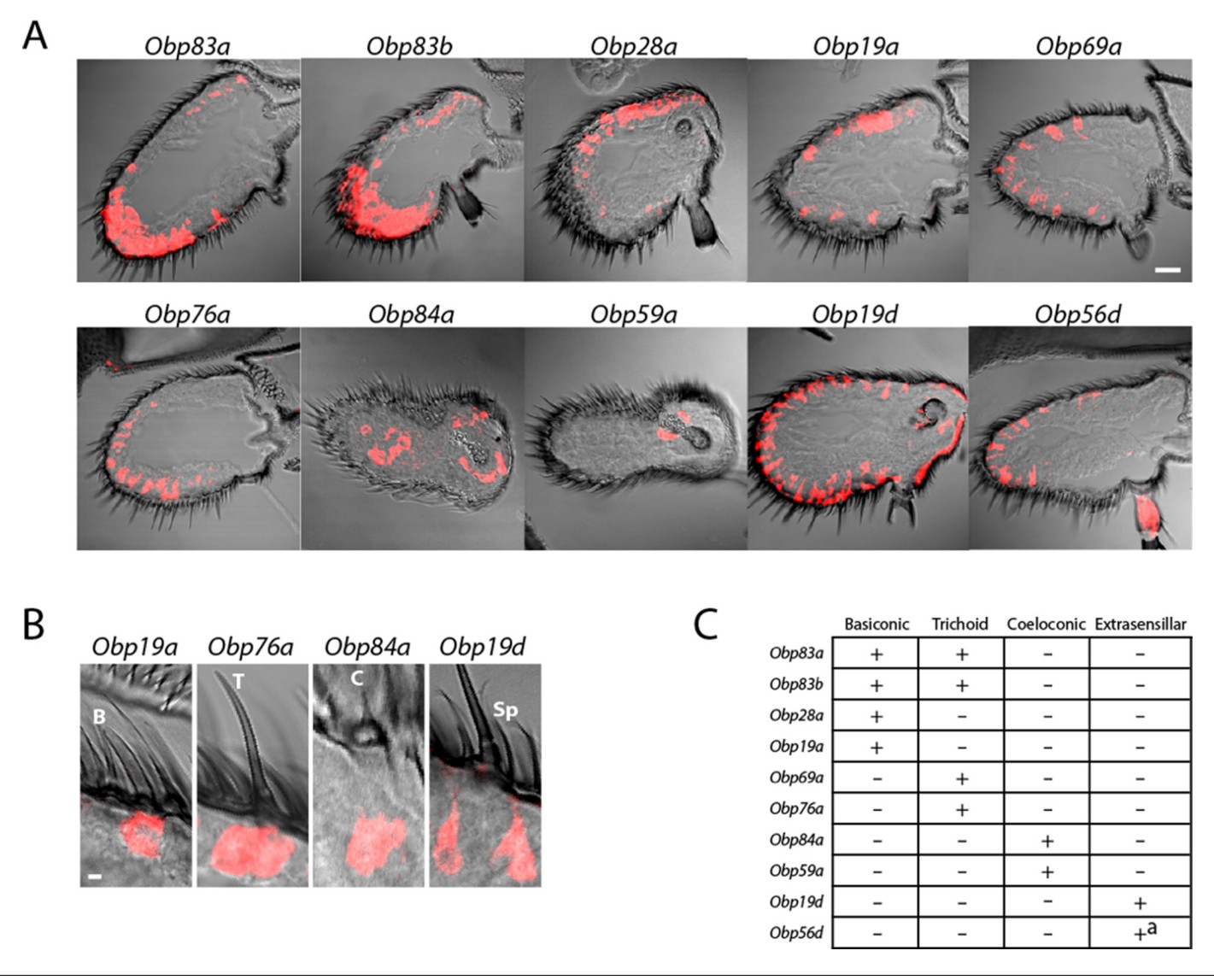

**Figure 2.** Diverse expression patterns of abundant *Obps* in the antenna. (**A**) In situ hybridization of *Obp* antisense RNA probes to antennal sections. Scale bar = 20 μm. Male and female antennae were examined with each probe and no sexual dimorphism was observed except that females appeared to show stronger labeling than males with *Obp56d* in the region where trichoid sensilla are found. (**B**) Higher magnification images of sensilla labeled with *Obp* probes: *Obp19a* in a basiconic sensillum (**B**), *Obp76a* in a trichoid sensillum (**T**), *Obp84a* in a coeloconic sensillum (**C**), and *Obp19d* in cells that are located between sensilla and that are associated with uninnervated spinules (Sp). Scale bar = 2 μm. (**C**) Summary of *Obp* expression patterns. +[a] indicates expression in the arista as well as in cells between sensilla of the third antennal segment.

ab8, and ab9) did not express detectable levels of *Obp83a*. The *Obp83b* gene, which lies less than 1 kb from *Obp83a* and encodes a protein with 68% amino acid identity to Obp83a (*Hekmat-Scafe et al., 1997*), maps to the same set of basiconic sensilla. By contrast, *Obp19a* maps to a different subset of sensilla. There are sensilla that express *Obp83a* and *b* but not *Obp19a*, sensilla that express *Obp19a* but not *Obp83a* or *b*, sensilla that express all, and sensilla that express none (*Figure 3B*). Interestingly, *Obp28a* is expressed in all 10 types of basiconic sensilla, suggesting that it could play a broad role in odor coding.

In summary, different *Obps* map to different functional subsets of basiconic sensilla, and basiconic sensilla express distinct subsets of *Obps*. A conclusion of particular interest is that ab8, which has

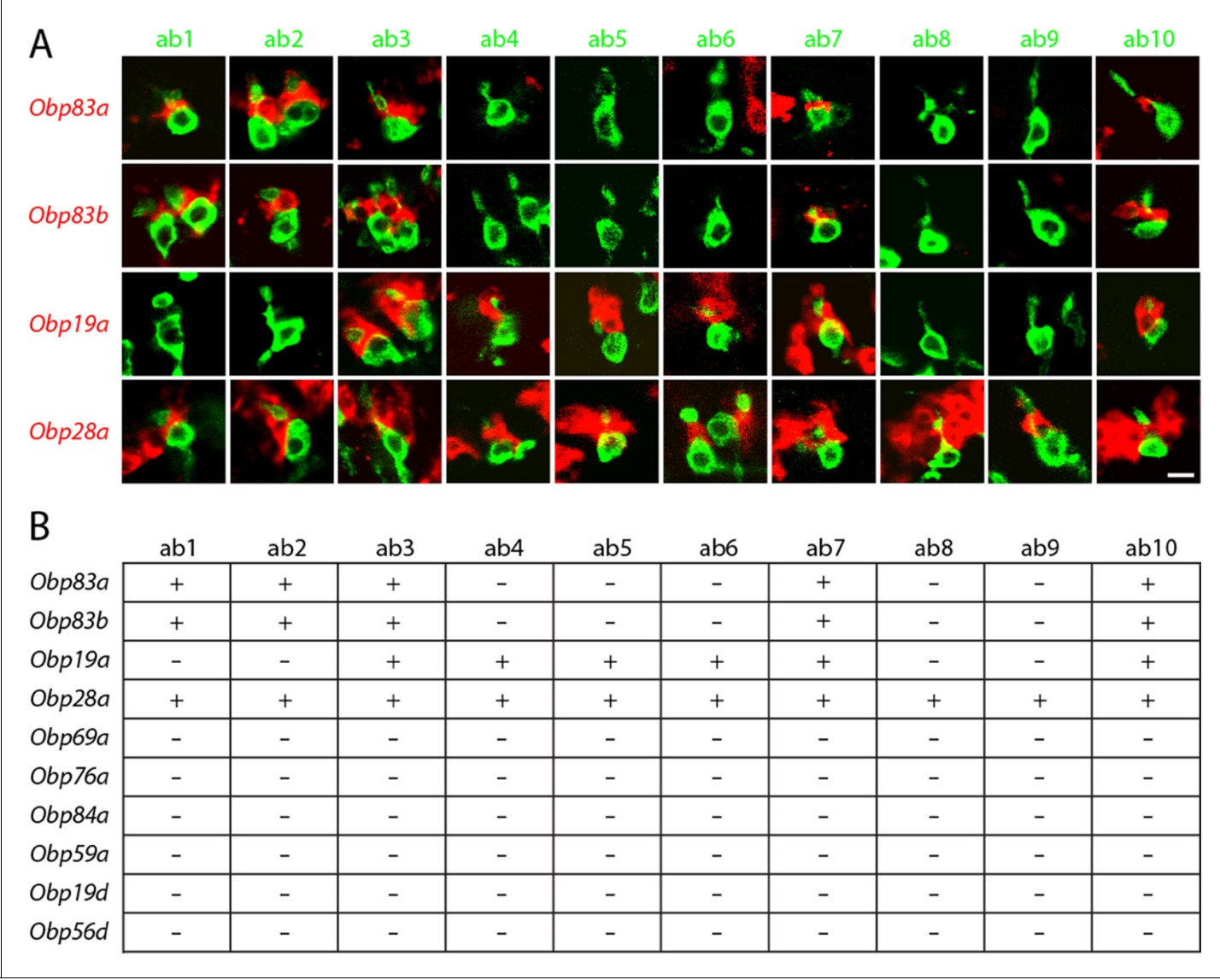

**Figure 3.** *Obps* are differentially expressed within basiconic sensillum types. (**A**) Confocal images of in situ hybridization to antennal sections labeled with antisense probes for the *Obps* (red) and an antibody against GFP (green) driven by *Or42b-GAL4* (ab1), *Or59b-GAL4* (ab2), *Or22a-GAL4* (ab3), *Or56a-GAL4* (ab4), *Or82a-GAL4* (ab5), *Or49b-GAL4* (ab6), *Or67c-GAL4; Or98a-GAL4* (used together to label ab7), *Or43b-GAL4* (ab8), *Or 67b-GAL4* (ab9), and *Or49a-GAL4* (**ab10**). Many panels show more than one sensillum. Thus in some panels, such as Obp83a/ab2, two sensilla are labeled by both the *Or-GAL4* driver and the *Obp* probe. In some other panels, such as Obp28a/ab1, *Obp* probes label multiple neighboring sensilla of which some are unlabeled by the *GAL4* driver. Scale bar = 5 μm. (**B**) Summary of *Obp* expression in ten basiconic types. An *Obp* is considered to be expressed in a sensillum type if it labeled a cell that wraps around the dendrites of ORNs in the majority of labeled sensilla examined. *Obp* expression was more difficult to identify with confidence in ab9 because of its proximity to other sensilla with strong *Obp* expression.

been well characterized (*Elmore et al., 2003*), expresses a single abundant *Obp*, as considered further below.

## *Obps* map to different cell types

We next examined *Obp* expression at still higher resolution: at the level of individual cell types. Olfactory sensilla contain not only neurons, but also thecogen, tormogen, and trichogen cells (*Shanbhag et al., 2000*). We found no evidence for expression of any *Obps* in neurons: (i) we did not observe axons or dendrites in the cells labeled by any of the 10 *Obp* probes (*Figures 3* and *4*); (ii) none of the *Obp* in situ hybridization probes co-labeled the neurons labeled by any of the 10

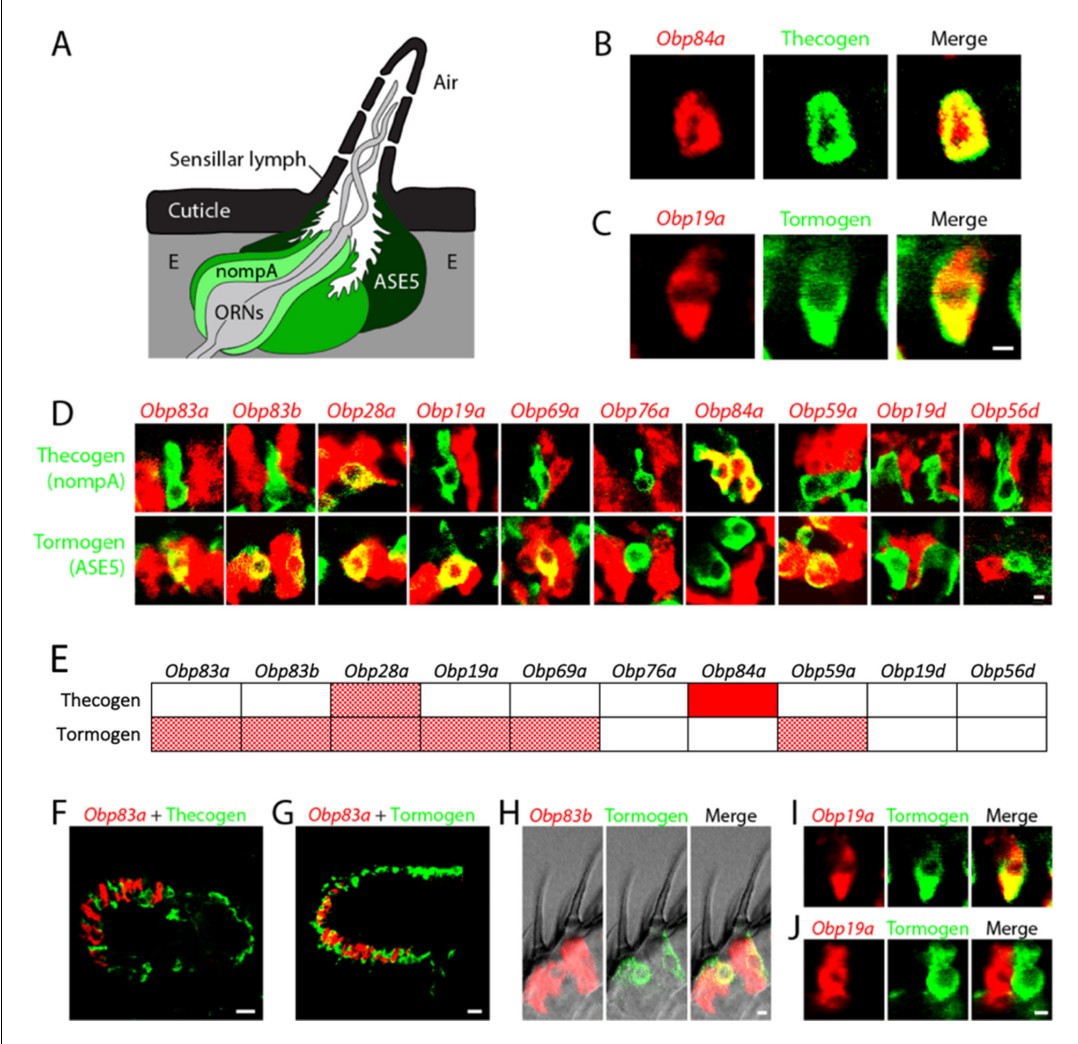

**Figure 4.** *Obps* are expressed in different cell types. (**A**) Diagram of a generic sensillum containing ORNs, thecogen cells labeled with *nompA-GAL4*, trichogen cells (Tr), and tormogen cells labeled with *ASE5-GAL4*, separated from neighboring sensilla by epidermal (E) cells. Adapted from (*Steinbrecht et al., 1992*). (**B–D**), (**F–J**) Confocal images of antennal sections labeled with *Obp* antisense probes (red) and an antibody against GFP (green) driven by the thecogen cell driver *nompA-GAL4* (**B,D,F**), and the tormogen cell driver *ASE5-GAL4* (**C–D,G–J**). Yellow indicates coexpression. (**E**) Summary of coexpression experiments. The dark, solid red rectangle indicates that *Obp84a* was co-expressed with the thecogen cell type marker sensilla consistently in those sensilla that express *Obp84a*. In many cases, indicated by light, stippled red rectangles, an *Obp* was co-expressed in a specific cell type in some but not all sensilla. An empty rectangle indicates that the *Obp* did not co-localize with that cell type marker in any sensilla examined. **F,G** Images of whole antennal sections. *Obp83a* is expressed in many cells that are not labeled by the thecogen (**F**) or tormogen cell driver (**G**). (**H**) Two trichoid sensilla that each house one tormogen cell (green) and two *Obp83b*-expressing cells (red). (**I**) *Obp19a* is coexpressed with the tormogen cell driver in one sensillum. The image shows the same cell as in (**C**) but in a different focal plane. (**J**) *Obp19a* is not coexpressed with the tormogen cell driver in another sensillum. Scale bars = 2 µm.

The following figure supplement is available for figure 4:

**Figure supplement 1.** Lack of *Obp* expression in ORNs.

*Or-GAL4* drivers in the double-labeling analysis (*Figure 3*); (iii) we carried out additional double-label experiments with *Obps* and *GAL4* drivers that label neurons in basiconic, trichoid, and coeloconic sensilla, and found no co-labeling (*Figure 4—figure supplement 1*).

We observed expression of *Obps* in thecogen and tormogen cells, as judged by double-label analysis using *Obp* probes and *GAL4* drivers that have previously been used as markers of these cell types in other tissues (*Barolo et al., 2000*; *Chung et al., 2001*; *Jeong et al., 2013*). For example, an

in situ hybridization probe for *Obp84a* labels cells that express a marker of thecogen cells, *nompA-GAL4* (*Figure 4A,B*), and an *Obp19a* probe labels cells that express a marker of tormogen cells, *ASE5-GAL4* (*Figure 4A,C*).

A systematic double-label analysis was carried out with all 10 *Obps* and the thecogen and tormogen cell markers (*Figure 4D*). Besides *Obp84a*, only one other *Obp*, *Obp28a*, was coexpressed with the thecogen marker, and this *Obp28a*-thecogen coexpression was observed in a very limited number of cells per antenna. Six of the *Obps* were coexpressed with the tormogen marker (*Figure 4D, E*). *Obp84a*, which showed strong labeling of thecogen cells, did not label tormogen cells. *Obp76a* did not label cells with either marker, suggesting that it is expressed in trichogen cells, a suggestion that we have been unable to confirm directly for lack of a suitable marker specific for trichogen cells in olfactory sensilla. Moreover, most of the *Obp* probes (including *Obp83a*, *Obp83b*, *Obp28a*, *Obp19a*, *Obp69a*, *Obp76a*) seem likely to be expressed in trichogen cells, based on the substantial number of cells they label in antenna sensilla that are not labeled by thecogen or tormogen markers; for example, *Obp83a* is not co-expressed with the thecogen marker (*Figure 4F*) and labels many cells that are not labeled by the tormogen marker (*Figure 4G*). Many individual sensilla contain more than one *Obp*-labeled cell, as can be seen in each of the two neighboring sensilla shown in *Figure 4H*. This finding is consistent with the interpretation that some *Obps* label more than one cell type (*Shanbhag et al., 2001a*; *Steinbrecht et al., 1992*). *Obp19d* and *Obp56d*, which label cells located between sensilla, do not co-localize with either marker (*Figure 4D,E*), consistent with the interpretation that they are expressed in epidermal cells.

Finally, we note with interest that *Obp19a* labeled the same cell as the tormogen marker in some sensilla (*Figure 4I*) but a different cell in other sensilla (*Figure 4J*). A simple interpretation of this result is that *Obp19a* is expressed in different cell types in different basiconic sensilla.

## Robust and increased responses to short odor pulses in the absence of an abundant Obp

The construction of an Obp-to-sensillum map provided an unprecedented opportunity to investigate Obp function in an incisive way. Of particular interest, the map shows that some sensilla express a single abundant *Obp*, *Obp28a* (*Figure 3B*). We sought to remove *Obp28a* genetically, with the goal of examining olfactory physiology in a sensillum whose Obp content had been drastically reduced if not eliminated.

Accordingly, we created a CRISPR/Cas9-mediated deletion of *Obp28a* (*Figure 5—figure supplement 1* and *Supplementary file 1*) and then outcrossed the deletion five times to a control genetic background. The mutation was verified by PCR analysis both before and after outcrossing, and we further confirmed the loss of *Obp28a* RNA by qPCR (not shown). We then analyzed the olfactory response of mutant ab8 sensilla via single-unit electrophysiology to assess the effect of removing its only abundant Obp.

Since Obps have been proposed to be required for the transport of hydrophobic odorants through the aqueous sensillum lymph, we first examined the effect of removing *Obp28a* on the detection of 1-octanol, an odorant that is highly hydrophobic (logP = 3.07). This odorant is found in citrus fruits and other plants, and it elicits modest responses from the receptors of both ab8A and ab8B neurons. Due to the difficulty of sorting spikes from the two ab8 neurons, we quantified the total number of spikes following stimulation.

We were surprised to find that the mutant sensillum responded robustly to the odorant across a broad concentration range, despite the lack of its single abundant Obp (red line in *Figure 5A*). The simplest interpretation of this result is that this sensillum can maintain a strong olfactory response with little if any Obp. Moreover, we were surprised to find that not only was the response robust, but that it was greater in the *Obp* mutant than in the control, over a broad concentration range.

To investigate the effect of Obp depletion in more detail, we examined the dynamics of olfactory response. Rather than focusing on the total number of spikes in a single 0.5 s interval, as in *Figure 5A*, we examined the numbers of spikes in 50 ms intervals and plotted the results as a peri-stimulus time histogram (PSTH)(*Figure 5B*). Again we observed a robust response in the mutant. The shape of the response was affected by Obp depletion, with the greatest effect occurring during the initial phase of the response. The peak response of the mutant was greater than that of the control across a broad range of 1-octanol concentrations. We note that the baseline firing rate was the same in the *Obp* mutant and the control (*Figure 5B*, 500–1000 ms).

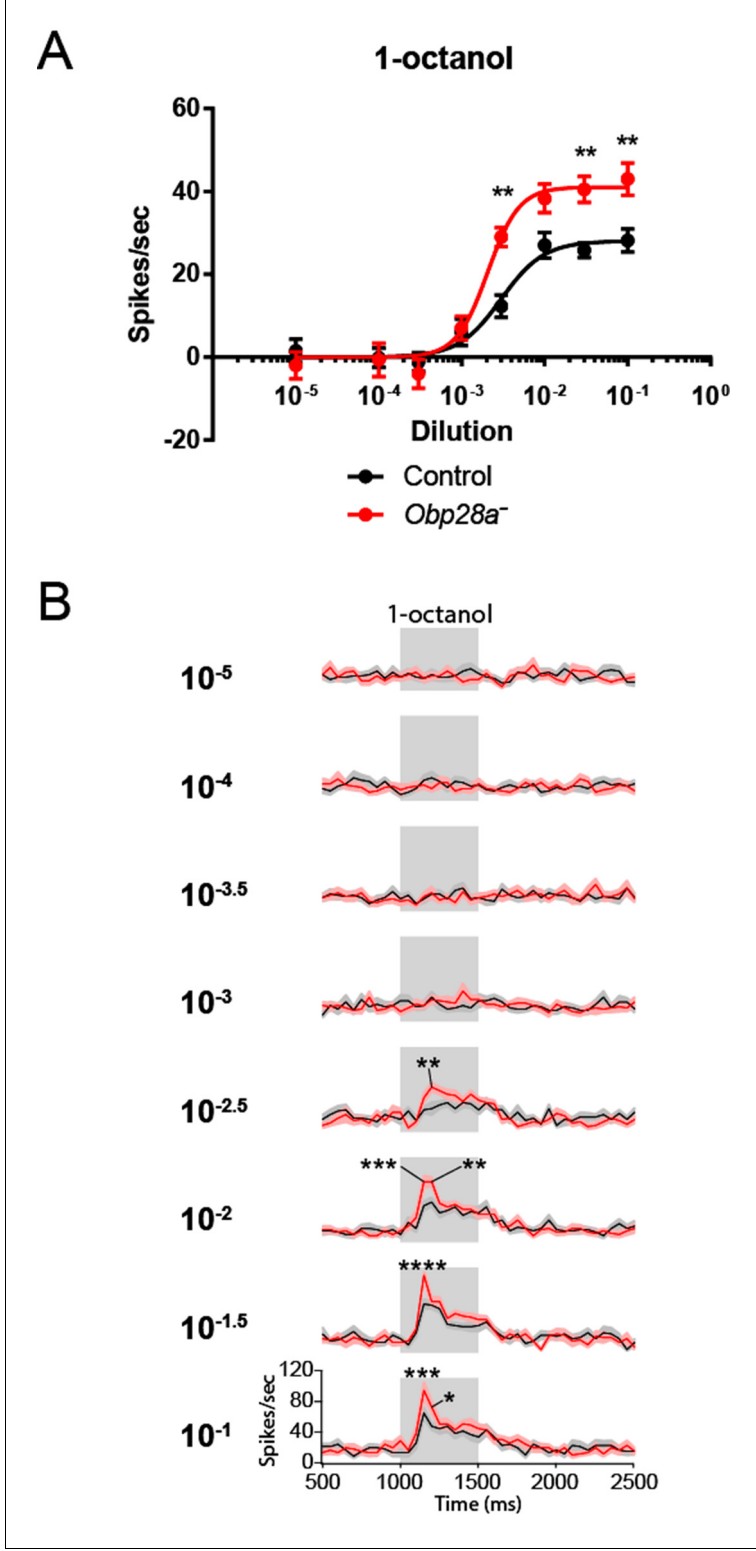

**Figure 5.** Robust and increased response of ab8 to 1-octanol in an *Obp28a* mutant. (**A**) Dose-response curves of control (black) and *Obp28a*⁻ (red) ab8 neuronal responses to a 0.5s pulse of 1-octanol. Responses were quantified by subtracting the spontaneous firing rate from the rate during the stimulus. The spike rates of both neurons were summed. (**B**) Peri-stimulus time histograms (PSTHs) of ab8 responses to increasing doses of 1-octanol. Gray boxes denote 0.5s stimulus presentations. The baseline firing rates of the mutant and control are comparable. Graphs

*Figure 5 continued on next page*

*Figure 5 continued*

display 2s time windows in 50 ms bins. Shaded areas surrounding each curve indicate SEM. * p<0.05, ** p<0.01, *** p<0.001, **** p<0.0001, n = 12.
The following figure supplement is available for figure 5:

**Figure supplement 1.** CRISPR mutant cloning and verification primers.

## Robust response and abnormal recovery from long odor stimuli

The preceding analysis concerned a brief odor stimulus: 0.5 s. In nature, flies also experience sustained olfactory stimuli. For example, flies spend prolonged periods of time in direct contact with food sources, which are intense sources of odor. We were interested in the possibility that Obp28a might be essential for response to such prolonged stimuli or for the recovery therefrom. The extremely high levels of *Obp28a* expression might have evolved to enhance odor coding under extreme conditions of olfactory stimulation.

We delivered a strong 1-octanol stimulus for 30 s. Consistent with our earlier results, the initial response of the mutant was greater than that of the control immediately following odor onset (*Figure 6A*). During the ensuing long stimulus period, the response of the mutant was comparable to that of the control, supporting the notion that the ab8 sensillum is capable of a robust olfactory response to a prolonged stimulus in the absence of an abundant Obp.

We then terminated the odor stimulus and measured the response. If Obp28a played an essential role in clearing high levels of odorant from the sensillum lymph, one might expect the $Obp28a^+$ control to show a faster decline in response than its $Obp28a^-$ counterpart. The opposite result was observed (*Figure 6A*). These findings argue against the possibility that Obp28a plays an essential role in clearing 1-octanol from the sensillum lymph after intense stimulation.

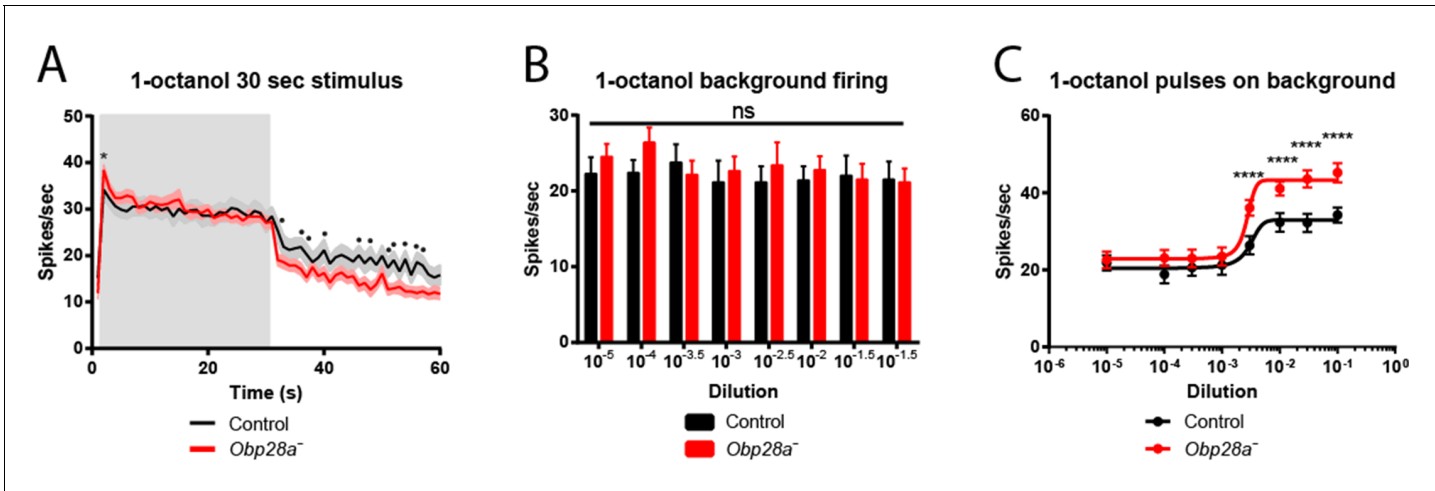

**Figure 6.** Altered responses of *Obp28a* in prolonged stimulus paradigms. (**A**) PSTH of ab8 responses to 30s presentations of a $10^{-1.5}$ dilution of 1-octanol in control (black) and *Obp28a⁻* (red). Gray box indicates stimulus presentation. Shaded areas display SEM. The graph displays a 60 s time window in 1 s bins. • p<0.05, n = 18; we note that for five of the bins, p<0.0001. We also asked whether the increased initial response of the mutant, indicated by *, was significant when the responses were examined in 50 ms bins, as in *Figure 5*, and found that p<0.05. (**B**) Background firing rates in response to prolonged (>5 min) stimulation of a $10^{-2.5}$ dilution of 1-octanol. Each bar represents data prior to the administration of a short pulse of the indicated dose. ns = not significant, n = 16. **C** Dose-response curves of ab8 neurons to increasing doses of 1-octanol superimposed on the $10^{-2.5}$ 1-octanol background. Spike rates are calculated from the number of spikes during the stimulus period, without subtracting the background firing rate. **** p<0.0001, n = 16.

## Robust and increased responses to short stimuli superimposed upon long stimuli

Having thus analyzed responses to both short and long odor stimuli, we then tested the response to short stimuli superimposed upon long stimuli. In nature, the olfactory system is often faced with the challenge of detecting an olfactory signal against a high background of odorant, for example in assessing local variation in the quality of a food source. Accordingly, we delivered a sustained background of 1-octanol and measured the response to a pulse of 1-octanol that was superimposed upon this background. More specifically, we kept the background at a constant level, and measured the responses to a series of superimposed 1-octanol pulses of increasing intensity.

Three results were obtained. First, prior to the odor pulses, the background firing rate was the same in the mutant as in the control (*Figure 6B*), consistent with the equivalent firing rates of mutant and control during the prolonged stimulus applied in *Figure 6A*. Second, the mutant responded robustly to superimposed pulses of odor, with a threshold between $10^{-3}$ and $10^{-2}$ dilutions (*Figure 6C*). Third, the magnitude of the mutant firing level in response to a pulse was greater than that of the control at all doses above the threshold, consistent with our earlier findings with pulses delivered in the absence of a background stimulus (*Figure 5*).

## Robust and increased responses to diverse odorants

The preceding analysis used a highly hydrophobic odor that elicited modest responses (<50 spikes/ s) even at high doses. We wanted to determine whether the depletion of an Obp would affect: (i) responses to a more hydrophilic odorant; (ii) strong responses; (iii) responses of both the ab8A and ab8B neurons. Although extensive screening (n>170 odorants) did not identify a highly hydrophobic odorant that strongly activated ab8A or ab8B (not shown), the more hydrophilic odorants ethyl acetate (logP = 0.73) and butyric acid (logP = 0.79) elicit strong responses (n>100 spikes/s) from the receptors of ab8A and ab8B, respectively (*Hallem and Carlson, 2006*).

We tested the response of the *Obp28a* mutant to 0.5 s pulses of butyric acid and ethyl acetate and found robust responses over a broad concentration range for both odorants (red lines in *Figure 7A and B*). The mutant responded more strongly than the control to each odorant at one or more concentrations. Butyric acid elicited a stronger response from the mutant over a wide range of concentrations (*Figure 7A*), and ethyl acetate elicited a stronger response from the mutant at one dose in the middle of its dynamic range (*Figure 7B*).

PSTH analysis revealed higher peak responses in the mutant for both butyric acid and ethyl acetate across a wide range of doses (*Figure 7C,D*). Following the termination of these short, 0.5s stimuli, decay dynamics appeared comparable in the two genotypes for butyric acid, but were faster in the mutant following stimulation with the highest ethyl acetate doses, arguing against a model in which the Obp is essential for clearing these odorants after such pulses.

We then tested a diverse panel of other odorants to ask whether the loss of an abundant Obp affected the response profile of the ORNs in ab8. *A priori*, Obp28a could differentially expedite the transport of a subset of odorants, or perhaps selectively filter some odorants so as to reduce their access to the dendrites in the sensillum. We selected a panel of eight odorants representing eight chemical classes—a sulfur compound, a lactone, a terpene, an aromatic, an aldehyde, a ketone, an alcohol and an ester—that elicit responses from the odor receptors of ab8 neurons (*Hallem and Carlson, 2006*). The odorants were tested at $10^{-2}$ dilutions, and also at $10^{-3}$ dilutions in the cases of the odorants that elicited the strongest responses. Each was delivered as a 0.5s pulse and the responses of both neurons were summed.

The response profiles appeared very similar in *Obp28a⁻* and *Obp28a⁺* (*Figure 7E*). The mean responses varied over a broad range, from 50 spikes/s to >200 spikes/s, and for both genotypes the lowest mean response was to 3-methylthio-1-propanol and the greatest response was to ethyl-3-hydroxybutyrate. In between these extremes, the rank order of stimuli was similar. The *Obp28a* mutant showed greater responses than the control to a subset of the compounds tested. In no case was the response of the mutant lower than that of the control.

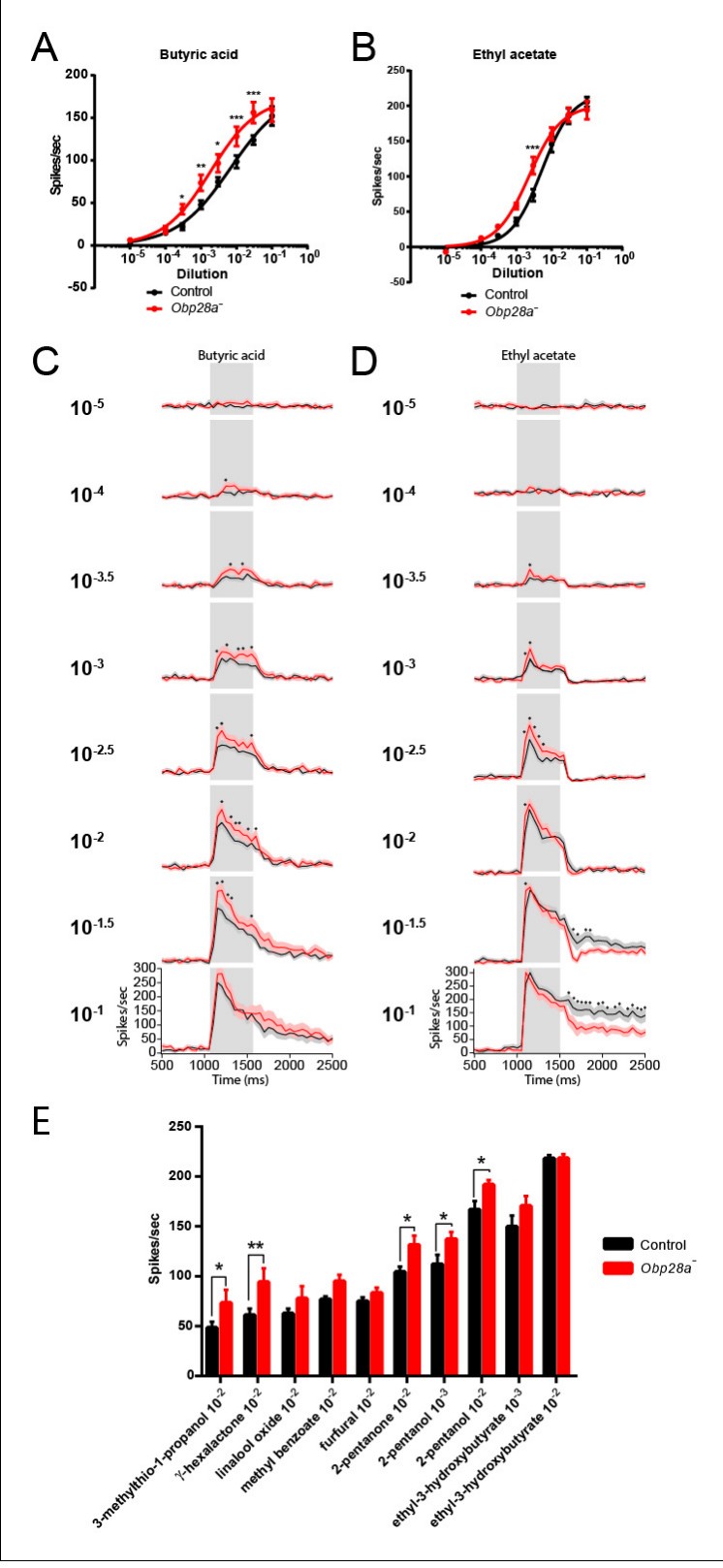

**Figure 7.** *Obp28a* mutants show robust and increased responses to other odorants that activate ab8A or ab8B
Dose-response curves of control (black) and *Obp28a⁻* (red) ab8 neuronal responses to a 0.5s pulse of butyric acid
(A) and ethyl acetate (B). Responses were quantified by subtracting the spontaneous firing rate from the rate
during the stimulus. * p<0.05, ** p<0.01, *** p<0.001. PSTHs of butyric acid (C) and ethyl acetate (D) responses are
*Figure 7 continued on next page*

*Figure 7 continued*

shown, with shaded areas indicating SEM. Gray boxes indicate 0.5 s stimulus presentations. Graphs depict 2s time windows with spikes summed in 50 ms bins. • indicates p<0.05 in all cases and p<0.01 in 80% of these cases. n = 12 (A,C) and 13 (B,D). (E) Responses of control (black) and *Obp28a⁻* (red) ab8 neurons to 0.5s pulses of odorants of different chemical classes: 3-methylthio-1-propanol (sulfur compound), γ-hexalactone (lactone), linalool oxide (terpene), methyl benzoate (aromatic), furfural (aldehyde), 2-pentanone (ketone), 2-pentanol (alcohol), and ethyl-3-hydroxybutyrate (ester). Responses were quantified by subtracting the spontaneous firing rate from the rate during the stimulus. * p<0.05, ** p<0.01. n = 12.

The following figure supplement is available for figure 7:

**Figure supplement 1.** Robust responses from ab4 sensilla of *Obp28a* mutants to odorants that activate ab4A or ab4B.

## Discussion

We have analyzed the molecular organization of the abundant Obps in the *Drosophila* antenna. We have established an Obp-to-sensillum map that illustrates principles of olfactory system organization and that allows a defined physiological analysis of the function of one Obp, Obp28a.

### The molecular organization of Obps

Our expression analysis highlights the great diversity of *Drosophila Obps*. The 10 abundant *Obps* are expressed in diverse antennal regions and in different morphological classes of sensilla. Within a morphological class, they are expressed in distinct functional types, and within different cell types. The diversity of these genes in expression pattern and amino acid sequence, together with their remarkably high abundance, provokes questions about their roles in olfactory function. The Obp-to-sensillum map lays a foundation for addressing these questions.

Most of these *Obps* are expressed in only one morphological class of sensillum. It seems likely that many Obps have evolved to fill specialized needs of particular sensilla. For example, coeloconic sensilla express a subset of *Obps* that does not overlap with those expressed by other sensilla. This pattern of mutually exclusive expression may reflect the ancient origin of the coeloconic sensilla, their unique double-walled architecture (*Shanbhag et al., 1999*), their response to many polar odor- ants (*Silbering et al., 2011*), or their expression of a different family of receptors, IRs (*Abuin et al., 2011*; *Benton et al., 2009*; *Croset et al., 2010*). The Obp-to-sensillum map now provides a founda- tion for designing ectopic expression experiments in which the function of an individual Obp can be examined in different sensillum types.

All 10 of the basiconic sensillum types express at least one of the abundant *Obps*. Most if not all of the coeloconic and trichoid sensilla also express at least one abundant *Obp* (*Hekmat-Scafe et al., 1997*; *Pikielny et al., 1994*; *Shanbhag et al., 2001*). These results are consistent with the concept that Obps are essential to the coding of olfactory information within sensilla.

The map reveals that some functionally distinct basiconic sensilla, such as ab1 and ab2, contain the same subset of abundant *Obps*. This finding supports the notion that Obps do not dictate the response profile of ORNs; rather, it is consistent with the conclusion from 'empty neuron' and heter- ologous expression analysis that the odor response profile of an ORN is conferred by the odor receptor that it expresses (*Abuin et al., 2011*; *Benton et al., 2009*; *Dobritsa et al., 2003*; *Hallem et al., 2004*; *Silbering et al., 2011*). When Ors from eight functional types of basiconic sen- silla, ab1-ab8, were individually expressed in the empty neuron, all yielded response profiles that agreed well with those observed in their endogenous neurons (*Hallem et al., 2004*). The map reveals that the sensillum used in the empty neuron expression system, ab3, expresses all four of the abundant *Obps* expressed in basiconic sensilla. Thus any abundant Obp that might be essential to the response of one of the 10 basiconic sensilla would be present in the ab3 test system.

We note that the Obp-to-sensillum map also invites analysis of the regulatory mechanisms by which it is established. A receptor-to-neuron map allowed incisive investigation of mechanisms by which individual ORNs select, from among 60 *Or* genes, which to express (*Barish and Volkan, 2015*; *Ray et al., 2007*, *2008*). Likewise, it should now be possible to elucidate mechanisms by which

individual sensilla select which *Obps* to express, for example by comparing regulatory regions of co-expressed *Obps*.

## Responses of a sensillum in vivo in the absence of an abundant Obp

We were surprised to find that elimination of the sole abundant Obp from a sensillum did not reduce the magnitude of its response to the tested odorants. We tested odorants of widely varying hydrophobicity and chemical class, including odorants that activate each of the ORNs in the sensillum. The results do not support the widespread belief that Obps are essential for the transport of odorants to receptors within all sensilla (*Figure 8A*); rather, the simplest interpretation of our results is that ab8 does not require an Obp to transport odorants to receptors.

How could a hydrophobic odorant traverse the aqueous sensillum lymph of ab8, if not via an Obp? Many sensilla of widely diverse insects contain tubular structures called pore tubules (*Steinbrecht, 1997*) (*Figure 8B,C*). These structures, which can be up to 1 µm long and 15–20 nm in diameter, extend from the wall pores of the sensilla into the interior of the sensillum, often making contact with dendritic membranes. The number of pore tubules per pore ranges up to 20, and the estimated number of pore tubules per sensillum ranges as high as 83,000 (*Steinbrecht, 1997*). In *Drosophila*, pore tubules have been observed in basiconic sensilla, but not in trichoid sensilla (*Shanbhag et al., 1999*), where Obp76a is expressed.

Pore tubules have been proposed as a conduit for the transport of hydrophobic odorants from the wall pores to the dendritic membranes. Thus an odorant would follow a three-dimensional trajectory through the air to the antenna, a two-dimensional path in the hydrophobic antennal surface cuticle to a wall pore, and then a one-dimensional trajectory toward the dendritic membrane via a pore tubule (*Figure 8B*) (*Adam and Delbruck, 1968*; *Kaissling et al., 1987*; *Steinbrecht, 1997*). There is evidence that pore tubules consist largely of lipids and proteins (*Keil, 1982*), likely allowing for diffusion of a hydrophobic odorant.

After the discovery of Obps, the view that Obps transport odorants to receptors was proposed and has prevailed in the field for over 30 years. Our results support the possibility that in some sensilla, pore tubules provide an alternative mechanism. We acknowledge that the following formal possibilities can not be excluded:

- deletion of *Obp28a* may result in a massive upregulation of another Obp. However, we tested this possibility by qPCR analysis and found no change (p>0.05) in the levels of four other *Obp* transcripts (*Obp19a*, *Obp19d*, *Obp83a*, and *Obp83b*, not shown).
- Obp28a may be essential for response to highly hydrophobic odorants that elicit very strong responses from neurons in

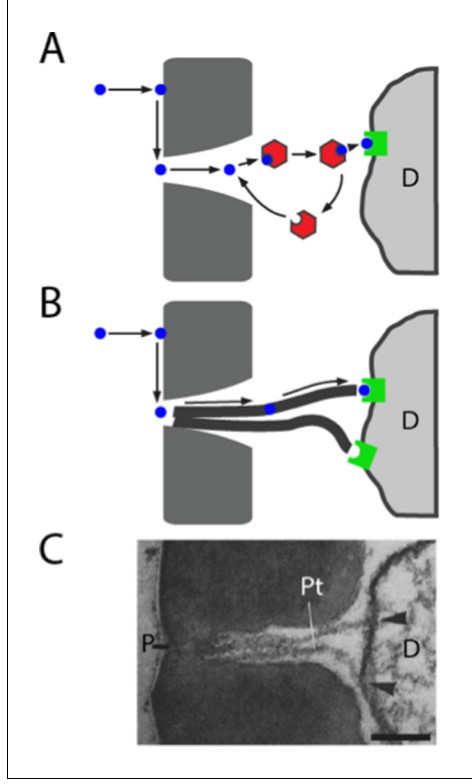

**Figure 8.** Models of odorant transport via Obps or pore tubules. (**A**) Odorant transport via Obps in a sensillum. An odorant molecule contacts the membrane, diffuses in the surface of the cuticle (dark gray) until it reaches a pore, enters the sensillum lymph through the pore, binds to an Obp (red hexagon), and is transported to an olfactory receptor (green square) on an ORN dendrite (**D**). (**B**) Odorant transport via pore tubules. An odorant molecule contacts and diffuses on the cuticle surface, enters a pore, and is transported along a pore tubule to an olfactory receptor on a dendrite (**D**). An Obp could bind to the odorant and affect dynamics at any point after the odorant reaches the pore. (**C**) Transmission electron micrograph of the pore (P) and pore tubules (Pt) of a trichoid sensillum of *Bombyx mori*. Adapted from (*Steinbrecht, 1973*). Pore tubules can be seen to contact an ORN dendrite (**D**) in two locations (arrowheads). Scale bar = 0.1 µm.

ab8. However, we have not been able to test this possibility because a screen for such odorants (n > 170) was unsuccessful.

- the ab8 sensillum may be atypical. However, a limited analysis of ab4, which expresses *Obp28a* and one other abundant *Obp*, *Obp19a*, revealed responses of comparable magnitudes in *Obp28a*⁻ and in an *Obp28a*⁺ control (13 odorants; *Figure 7—figure supplement 1*).
- the function of Obp28a in ab8 could be redundant with that of another Obp that is expressed at drastically lower levels in ab8. However, the most abundant unmapped *Obp* is expressed at <5% the level of *Obp28a* (*Menuz et al., 2014*). Moreover, we did not detect labeling of ab8 by in situ hybridization with this *Obp*, nor with the following three in descending rank order, nor with any of four other *Obps* tested (*Figure 1D*, *Figure 1—figure supplement 1*; the *Obps* tested were *Obp57c*, *Obp57b*, *Obp83ef*, *Obp19b*, *Obp44a*, *Obp57a*, *Obp58b*, *Obp83cd*, not shown). We note finally that abundance has been quantified in terms of RNA levels, which provide an imperfect reflection of protein levels.

A second surprise was that the response to an odor pulse was greater in the mutant, for a number of odorants. The mutant also showed an accelerated decline in the firing level following the termination of a prolonged odor stimulus. The overall response profile to a panel of diverse odorants, however, was similar between the mutant and control. These results are interesting because they argue against some alternative models of Obp28a function. Specifically, in addition to transporting odorants, Obps have been proposed to protect odorants from degradative enzymes, to inactivate odorants following stimulus termination, or to filter them (*Kaissling, 2001*; *Leal, 2013*; *Pelosi and Maida, 1995*). If Obp28a protected odorants in ab8 we would expect a greater initial response in the control than in *Obp28a* mutants. If Obp28a inactivated odorants at the end of the odor response, we would expect a faster decline in the control at the end of odor stimulation. If Obp28a selectively filtered certain odorants, we would expect differences in the response profiles of mutant and control. We note moreover that the similarity in response profiles is consistent with results from the empty neuron system. If Obp28a selectively filtered certain odorants, we would expect the Or35a receptor to confer a different odor response profile in basiconic sensilla, which express *Obp28a*, and in coeloconic sensilla, which do not (*Figure 2C*). By contrast, the response profiles of Or35a in the two sensillum types are very similar (*Hallem et al., 2004*).

We do not claim that the role of Obp28a in ab8 can be extrapolated to all Obps and all olfactory sensilla. Obp28a is expressed in all basiconic sensilla, and could play a broader role than Obps expressed in only a subset of sensilla. One of the most intriguing aspects of Obps and sensilla is their diversity. Different Obps are likely to have different functions: there is evidence for reduction in olfactory function following reduction in the levels of certain Obps (*Biessmann et al., 2010*; *Pelletier et al., 2010*; *Swarup et al., 2011*; *Xu et al., 2005*), and another Obp has been found to contribute to the inhibition of neurons in the taste system (*Jeong et al., 2013*). In this regard, we found that the most abundant Obp transcript in the antenna, *Obp19d*, is expressed in epidermal cells and not accessory cells, consistent with the findings of (*Park et al., 2000*), and is thus unlikely to be secreted into sensillar lymph. It seems plausible that Obp19a binds ligands other than odorants, as may other Obps that are expressed outside the olfactory system (*Arya et al., 2010*; *Li et al., 2005*; *Pitts et al., 2014*). Not only are Obps highly diverse, but sensilla also are diverse in structure and function, as illustrated by the lack of visible pore tubules in trichoid sensilla of *Drosophila* (*Shanbhag et al., 1999*). We have focused on a single Obp and a single sensillum type because the map constructed in this study allowed us to manipulate them with unparalleled definition.

How does the presence of Obp28a reduce the initial response to odorant and prolong the response following termination of a long, intense stimulus? It is possible that following the sudden influx of an odor stimulus, some of the odorant binds to Obp28a, thereby reducing the amount available for receptor activation. After the termination of a prolonged stimulus, odorant released from the Obp-odorant complex might increase the level available for receptor binding. In both cases the Obp would buffer the system against sudden changes in odor level.

The olfactory circuit has evolved a cellular mechanism of gain control, in which local interneurons inhibit ORNs (*Olsen and Wilson, 2008*; *Root et al., 2008*). This gain control is believed to play a critical role in preventing network saturation and in improving odorant discrimination. It is possible that Obp28a provides an additional mechanism of gain control, which acts at the molecular level as

opposed to the cellular level. This molecular form of gain control would precede the cellular form, occurring even before the odor receptor or the neuron were activated.

We note finally that even small effects on ORN firing rate can have large effects on odor perception and behavior. ORNs form synapses with second-order neurons called projection neurons (PNs), and the relationship between the firing rates of ORNs and PNs is non-linear (*Olsen and Wilson, 2008*). Moreover, PNs also appear to respond most vigorously to odor onset (*Olsen and Wilson, 2008*; *Wilson, 2013*). Thus, even a modest alteration in the firing rate during the initial phase of response could have an important effect on the way olfactory information is represented in the CNS and on the behavior that it drives.

# Materials and methods

## *Drosophila* stocks

*Or-GAL4* drivers were from the Bloomington *Drosophila* Stock Center (NIH P40OD018537). *nompA-GAL4* and *ASE5-GAL4* were provided by C. Montell and described previously (*Barolo et al., 2000*; *Chung et al., 2001*). GAL4 drivers were crossed to a *UAS-mCD8:GFP* line (*Lee and Luo, 1999*). *white Canton-S (wCS)* and *Obp28a⁻* flies (described below) were used for electrophysiology experiments. *Obp28a⁻* was outcrossed for five generations to *wCS* to reduce the risk of background effects.

## Transgenic fly generation
### Guide chiRNA cloning
Gibson Assembly was used to clone *pU6-BbsI-chiRNA* plasmids containing each of the chiRNAs (*Barnes, 1994*), following protocols for the Gibson Assembly Master Mix (New England BioLabs, Ipswich, MA). Primers for this technique contain extended regions which are digested by exonucleases to create single-stranded 3' overhangs, which are filled and sealed by polymerase and DNA ligase, respectively. While the reverse primer is complementary to the plasmid itself (Primer #1, *Figure 5—figure supplement 1*), the forward primers also include the 20-nt guide chiRNA sequence with the PAM sequence omitted (Primers #2 and 3, *Figure 5—figure supplement 1*). Primer sequences complementary to the plasmid are represented in *Figure 5—figure supplement 1* in lowercase letters. 20-nt guide chiRNAs were selected with the aid of the CRISPR Optimal Target Finder resource on the flyCRISPR website (*Gratz et al., 2014*). CRISPR targets with 5' G and NGG PAM sequences were selected for optimal U6-driven transcription and subsequent cleavage stringency. Selected cut sites were located 19-nt downstream of the 5' end, and 37-nt downstream of the 3' end, of the *Obp28a* coding sequence.

### Homology-driven repair template cloning
Homology arms extending 0.65 kb upstream and 1.02 kb downstream of the cut site were incorporated into multiple cloning sites of the pHD-DsRed-attP vector (*Gratz et al., 2014*), using amplification primers #4–7 indicated in *Figure 5—figure supplement 1*. To facilitate screening of *Obp28a* deletion mutants, this replacement donor plasmid contains a removable *LoxP*-flanked *DsRed*, which was thus inserted in the genome at the same locus. An *attP ΦC31* site was simultaneously inserted to facilitate future targeting of this locus.

### Embryo injection
300 $y^2$ $cho^2$ $v^1$; P(nos-Cas9, $y^+$, $v^+$)3A/TM6C, Sb Tb (CAS-0003) (*Kondo and Ueda, 2013*) embryos were injected with guide chiRNA and donor plasmids by Bestgene, Inc. (Chino Hills, CA). *CAS-0003* flies express *Cas9* under the control of *nos* regulatory sequences, inserted in the third chromosome. These flies were optimal for *Obp28a* gene deletion due to the intact second chromosome, as well as the visible and removable endogenous *Cas9* gene. Of the 150 surviving larvae,~60 G0 adults were crossed to $w^{1118}$. Two non-sibling G1 adults expressing *DsRed* were identified, and were individually backcrossed to *wCs* for five generations.

### PCR screening

Primer pairs extending beyond the *Obp28a* coding sequence (Primers #9 and 10, *Figure 5—figure supplement 1*) and homology arms (Primers #8 and 11, *Figure 5—figure supplement 1*) were used to verify the gene deletion, in two independent experiments, one before and one after backcrossing.

## RNA probe synthesis for in situ hybridization

The coding region of each *Obp* was amplified from *CS* antennal cDNA and cloned into the pGEM-T Easy vector (Invitrogen, Waltham, MA) for transcription. For genes with multiple transcripts, the primers were designed to encompass the region present in all versions. Plasmids were linearized with SpeI, NotI, or AatII (New England BioLabs). Digoxigenin (DIG) and Fluorescein (FITC) labeled probes were created using DIG RNA Labeling Kit SP6/T7 and Fluorescein-labeled UTP (Roche, Branford, CT), and purified with the RNEasy Cleanup Kit (QIAGEN, Germantown, MD).

## In situ hybridization and immunohistochemistry

We used seven day-old flies in all staining experiments except that we used flies less than 1-day-old, many within a few hours of eclosion, when labeling with *ASE5-Gal4* and *nompA-Gal4* because these markers are developmentally regulated and staining was weak in older flies. Male and female flies were anesthetized, placed in a collar, covered with OCT (Tissue-Tek, VWR, Radnor, PA), and frozen on dry ice. 14 µm antennal cryosections were collected on slides and stained as previously described (*Menuz et al., 2014*). In brief, sections were fixed and acetylated at room temperature, pre-hybridized, and then incubated with DIG and/or FITC probes at 65°C overnight. Detection of probes was carried out with anti-DIG-POD or anti-FITC-POD in 1% Blocking Reagent (Roche) and amplified with Cy3 or Cy5 TSA (Perkin Elmer, Waltham, MA). GFP was detected with mouse-anti-GFP (Roche) and Alexa-fluor-488 donkey-anti-mouse antibodies (Invitrogen). All microscopy was performed using a Carl Zeiss LSM 510 Laser Scanning Confocal Microscope and images were processed with ImageJ software.

## Electrophysiology

We used 6–8 day old female flies for single-sensillum recordings essentially as described previously (*Dobritsa et al., 2003*). Filtered AC signals (50–2000 Hz) were recorded and digitized with a Digidata 1440 digitizer and Axoscope 10.5 software (Molecular Devices). Action potentials were detected and counted using custom Matlab (MathWorks) scripts written by Carlotta Martelli (*Martelli et al., 2013*), with few if any modifications. All spikes were counted due to difficulty in reliably sorting A and B neurons in ab8 sensilla. Impulse responses are defined as the firing rate during the stimulus period minus the baseline firing rate. Odors typically take 100 ms to reach the antenna, due to the length of the odor delivery tube, so the stimulus period is considered to start 100 ms after the opening of the odor delivery valve and persists for the length of the pulse, which in nearly all cases was 500 ms. The baseline firing rate was calculated by counting the number of spikes during the 500 ms period prior to odorant release. To generate dose response curves, each sensillum was tested with all concentrations of odor in ascending order of dose, and no more than 3 sensilla were analyzed per fly. ab8 sensilla were identified by their location and physical attributes and confirmed by their strong response to $10^{-3}$ 2,3-butanediol (*Hallem and Carlson, 2006*). Spikes for peri-stimulus time histograms (PSTHs) were binned in 50 ms intervals except that in the 30s stimulus paradigm they were binned in 1s intervals. Statistical significance was assessed using Prism's (GraphPad) two-way repeated measures ANOVA followed by Bonferroni's post-hoc test. Values shown are the mean +/− SEM.

## Odor stimuli

Odor stimuli were prepared and delivered essentially as described previously (*Hallem et al., 2004*). 1-octanol, butyric acid, 3-methylthio-1-propanol, γ-hexalactone, furfural, 2-pentanone, 2-pentanol, ethyl-3-hydroxybutyrate, and were purchased from Sigma Aldrich (St. Louis, MO), methyl benzoate and linalool oxide were purchased from Fluka (St. Louis, MO), and ethyl acetate was purchased from J.T. Baker (Center Valley, PA). Odorants of the highest grade available were used and were then diluted in paraffin oil (Fluka). For 500 ms pulses, 50 µl of diluted odorant was applied to a 13 mm

filter paper disc inside a Pasteur pipette and capped with a 1 ml pipette tip to create an odor cartridge. Cartridges were allowed to equilibrate for 20 min before use. Each cartridge was used no more than three times and with more than 10 min between uses, to allow the odor to re-equilibrate. Mutant and control flies were tested in parallel, on the same day, and in the same manner. Stimuli were presented by placing the tip of the cartridge into a glass tube delivering a humidified air stream (~2000 ml/min) to the fly, and administering a 500 ms pulse of air (~200 ml/min) through the cartridge. For experiments with a 30 s odor stimulus, 25 ml pipettes were used in place of Pasteur pipettes, with three filter papers, each 55 mm in diameter, and 1 ml of odor dilution. These large cartridges were used up to eight times, since independent PID measurements showed no substantial decrease in odor concentration after more than 10 uses. For experiments with a background odor, a 125 ml flask with 5 ml of odor dilution was inserted between the main airstream and the odor delivery tube. Flies were exposed to the background odor for more than 5 min before beginning the experiment.

## Acknowledgements

We thank M Harrison, K O'Connor-Giles, and J Wildonger for plasmids pDsRed-attP (Addgene plasmid 51019; unpublished) and pU6-BbsI-chiRNA (Addgene plasmid 45946; Gratz et al., 2013). We thank T Emonet, C Martelli, M Demir, and S Gourur-Shandilya for discussion and help. We thank C Montell for fly lines. We thank T-W Koh, K Menuz, and C-Y Su for experimental advice, and F Marion-Poll, Z He, and F Vonhoff for comments on the manuscript. The work was supported by NSF Graduate Research Fellowships to NKL and JSS, by the Dwight N and Noyes D Clark Scholarship Fund and NIH T32 GM007499 (JSS), and by grants from the NIH to JRC.

## Additional information

### Funding

| Funder | Grant reference number | Author |
| --- | --- | --- |
| National Institute on Deafness and Other Communication Disorders | | John R Carlson |
| National Science Foundation | | Nikki K Larter<br>Jennifer S Sun |
| National Institutes of Health | T32 GM007499 | Jennifer S Sun |
| Dwight N. and Noyes D. Clark Scholarship Fund | | Jennifer S Sun |

The funders had no role in study design, data collection and interpretation, or the decision to submit the work for publication.

### Author contributions

NKL, JSS, Conception and design, Acquisition of data, Analysis and interpretation of data, Drafting or revising the article, Contributed unpublished essential data or reagents; JRC, Conception and design, Analysis and interpretation of data, Drafting or revising the article

### Author ORCIDs

Nikki K Larter, http://orcid.org/0000-0002-1938-1929
Jennifer S Sun, http://orcid.org/0000-0002-4274-0504
John R Carlson, http://orcid.org/0000-0002-0244-5180

## Additional files

**Supplementary files**
• Supplementary file 1. Deleted regions of *Obp28a* gene Sequence of *Obp28a* genomic region. PAM cleavage sites were located 19-nt downstream of the 5' end and 37-nt downstream of the 3' end of the *Obp28a* coding sequence.

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
