## [Decision Letter]

Thank you for submitting your article "Organization and function of *Drosophila* odorant binding proteins" for consideration by *eLife*. Your article has been favorably evaluated by a Senior Editor and two reviewers, one of whom is a member of our Board of Reviewing Editors.

As you note below, both reviewers consider the work significant and data of high quality. Both wish that the findings can be more generalized, with somewhat overlapping suggestions. After discussions between the reviewers, we agree that you should try your best in your revised manuscript to satisfy both reviewers' critiques (in particular reviewer #2, who has more substantial suggestions), which can be achieved in several months of time frame.

Reviewer #1:

Larter and colleagues provided a comprehensive characterization of 10 most abundantly expressed odorant binding proteins (obps), a large family of proteins expressed in insect olfactory systems previously hypothesized to function in carrying odors to the dendritic surface of olfactory receptor neurons (ORNs). They found that each is expressed in a specific type(s) of sensilla, in specific subtypes within the basiconic sensilla, and in specific cell types associated with ORNs (but not in ORNs themselves). By analyzing loss of function phenotypes of Obp28a that is uniquely expressed in ab8 basiconic sensillum, they provide strong evidence against the prevailing hypotheses about obp function: transporting odorants to ORNs or removing odorants efficiently after the stimulus period. Rather, their data suggest a new function for obp: gain control to buffer sudden changes in odorant concentrations.

This paper is well written and technical quality of the data is high. I recommend publication in *eLife* after revisions that address the following (relatively minor) points.

1) Although the authors were careful in qualifying their statements in Discussion based on detailed findings of a single ORN class, they should be more circumspect in their Abstract about the limitations of extrapolating the generality of their findings.

2) In point iii) of the fourth paragraph of the subsection “Responses of a sensillum in vivo in the absence of an abundant Obp”: the authors describe additional analysis for ab4 as data not shown. These should be shown as a supplement. It would have been stronger if the authors examined ab9, which only expresses Obp28a as an abundant Obp.

Reviewer #2:

In the manuscript by Larter, Sun, and Carlson, the authors investigate the function of odorant binding proteins (OBPs), which are among the most widely expressed, yet poorly understood, proteins found in olfactory organs (but also found elsewhere). The authors generate a detailed anatomical map for the top 10 antennal expressed (by RNA-seq) OBPs, finding that OBPs differ in which sensilla they are expressed, and also in which cells they are expressed within that sensilla. They identify Obp28a as being the main OBP expressed in ab8 sensilla, generate an *obp28a* mutant, and characterize odor response properties of the neurons within ab8. Surprisingly, they find that *obp28a* mutants exhibited increased responses, especially at higher odor concentrations, and often exhibited sustained activity after the odor pulse. This suggests that this OBP may function to sequester odorants, perhaps as a form of gain control. This highlights an unexpected role for OBPs in olfactory signaling.

The characterization of OBPs is important as they underlie a long-lasting puzzle- what is the function of these highly expressed olfactory proteins in odor responses? The anatomical descriptions of OBP expressions will be useful for future investigations. The manuscript is clearly written, and the data is high quality, well presented and described.

However, the manuscript does not go far enough in addressing the biological puzzle regarding OBPs. It helps clarify more about what an Obp does not do, but we are still left with questions on what they could do. The other most well characterized OBP, LUSH (Obp76a) has a very different function in at1 sensilla- that is to enhance signaling by the cVA pheromone. In the current work, the authors found Obp28a functions to reduce signaling by odorants. This is indeed interesting, as it suggests the roles for Obps will be many and various. But we are left wondering if Obp28a represents a single exception, or part of the varied landscape of Obp functions. As such, it would be greatly informative if the authors had examined the role of at least one additional Obp on odorant-induced activities. Detailed below is a possible experimental strategy, but other examples to address this would be acceptable.

The authors chose mutating Obp28a and examining ab8 as Obp28a is the only Obp expressed in ab8. This is a reasonable approach, but could be extended using methods routine to the group. This would yield greater insights into Obp functions that would fit well into the scope and aims of this manuscript. For example, ab4/ab5/ab6 sensilla express only Obp28a and Obp19a (Figure 3). Thus, by mutating (preferable) or using RNAi against Obp19a (available from Bloomington; could use tormogen-GAL4 or even tubulin-GAL4), the authors could extend their analyses to ab4/ab5/ab6 sensilla response profiles when experiments are performed in the *obp28a* mutant background. ab4 contains Or7a and Or56a neurons whose response profiles are well characterized, and could be very informative regarding Obp functions. For example, Or7a responds to a number of different odorants (for example E2-hexenal and benzaldehyde) whereas Or56a is the sole responder to geosmin. Does high activation of Or56a by geosmin require Obp functions, as seen with cVA and LUSH/Obp76a? This offers many opportunities to examine the role of a different Obp on neuronal firing properties. The authors may even have started these experiments, as suggested by data not shown in point iii) of the fourth paragraph of the subsection “Responses of a sensillum in vivo in the absence of an abundant Obp. Nonetheless, ab5 or ab6 sensilla could be targeted for analyses at the author's discretion in an *obp19a, obp28s* double mutant. It would also be informative to compare the odor-responses in ab4/ab5/ab6 when Obp19a is removed in a wild-type background (instead of *obp28a* mutant background), which might give insights into additional/different roles Obp28a may play in different sensilla and with a different set of odorants.

Alternative suggested experiments to more fully address function of Obps:

The authors could examine the odor response profile of ab9 neurons in the *Obp28a* mutant.

The authors could examine the effects on odor responses when Obps are ectopically expressed. For example, using the *obp28a* mutant background, Obps not expressed in ab8/ab9 could be introduced (e.g., Obp19a, Obp83b). Example genotype: ASE5-GAL4, UAS-Obp83a, *obp28a-/-.* This would address how different Obps may alter odor-induced activity.

---

## [Author Response]

*Reviewer #1:*

*[…] This paper is well written and technical quality of the data is high. I recommend publication in eLife after revisions that address the following (relatively minor) points.*

*1) Although the authors were careful in qualifying their statements in Discussion based on detailed findings of a single ORN class, they should be more circumspect in their Abstract about the limitations of extrapolating the generality of their findings.*

There is a tight word limit to the Abstract but we have made a change in wording to emphasize that we have examined a single Obp. Specifically, the Abstract now reads: “[…] The results suggest that this Obp is not required for odorant transport and that this sensillum does not require an abundant Obp. The results further suggest a novel role for this Obp […]”

*2) In point iii) of the fourth paragraph of the subsection “Responses of a sensillum* in vivo *in the absence of an abundant Obp”: the authors describe additional analysis for ab4 as data not shown. These should be shown as a supplement. It would have been stronger if the authors examined ab9, which only expresses Obp28a as an abundant Obp.*

As requested, we have added these data as Figure 7—figure supplement 1.

*Reviewer #2:*

*[…] However, the manuscript does not go far enough in addressing the biological puzzle regarding OBPs. It helps clarify more about what an Obp does not do, but we are still left with questions on what they could do. The other most well characterized OBP, LUSH (Obp76a) has a very different function in at1 sensilla- that is to enhance signaling by the cVA pheromone. In the current work, the authors found Obp28a functions to reduce signaling by odorants. This is indeed interesting, as it suggests the roles for Obps will be many and various. But we are left wondering if Obp28a represents a single exception, or part of the varied landscape of Obp functions. As such, it would be greatly informative if the authors had examined the role of at least one additional Obp on odorant-induced activities. Detailed below is a possible experimental strategy, but other examples to address this would be acceptable.*

*The authors chose mutating Obp28a and examining ab8 as Obp28a is the only Obp expressed in ab8. This is a reasonable approach, but could be extended using methods routine to the group. This would yield greater insights into Obp functions that would fit well into the scope and aims of this manuscript. For example, ab4/ab5/ab6 sensilla express only Obp28a and Obp19a (Figure 3). Thus, by mutating (preferable) or using RNAi against Obp19a (available from Bloomington; could use tormogen-GAL4 or even tubulin-GAL4), the authors could extend their analyses to ab4/ab5/ab6 sensilla response profiles when experiments are performed in the obp28a mutant background. ab4 contains Or7a and Or56a neurons whose response profiles are well characterized, and could be very informative regarding Obp functions. For example, Or7a responds to a number of different odorants (for example E2-hexenal and benzaldehyde) whereas Or56a is the sole responder to geosmin. Does high activation of Or56a by geosmin require Obp functions, as seen with cVA and LUSH/Obp76a? This offers many opportunities to examine the role of a different Obp on neuronal firing properties. The authors may even have started these experiments, as suggested by data not shown in point iii) of the fourth paragraph of the subsection “Responses of a sensillum* in vivo in the absence of an abundant Obp. Nonetheless, ab5 or ab6 sensilla could be targeted for analyses at the author's discretion in an obp19a, obp28s double mutant. It would also be informative to compare the odor-responses in ab4/ab5/ab6 when Obp19a is removed in a wild-type background (instead of obp28a mutant background), which might give insights into additional/different roles Obp28a may play in different sensilla and with a different set of odorants.

*Alternative suggested experiments to more fully address function of Obps:*

*The authors could examine the odor response profile of ab9 neurons in the Obp28a mutant.*

*The authors could examine the effects on odor responses when Obps are ectopically expressed. For example, using the obp28a mutant background, Obps not expressed in ab8/ab9 could be introduced (e.g., Obp19a, Obp83b). Example genotype: ASE5-GAL4, UAS-Obp83a, obp28a-/-. This would address how different Obps may alter odor-induced activity.*

We are in fact examining the role of another Obp but this work represents a major, independent study that will be submitted at a later time with a different first author.

Rather, we have extended the manuscript by adding a new figure (Figure 7—figure supplement 1), requested also by reviewer #1, that examines ab4, including the response of Or56a to geosmin, in the absence of Obp28a. The results show that in ab4, as in ab8, Obp28a function is not required for a robust odorant response to geosmin or other odorants. Specifically, we now show the responses of both ab4A and ab4B neurons to a panel of 13 odorants and the responses are comparable in mutant and wild type.

We agree that ectopic Obp expression would be very interesting, but generating the necessary constructs and introducing them into the mutant ab8 sensilla is a major undertaking that would take an extended period of time and we feel would be beyond the scope of this already quite extensive study.